# A²TG: Adaptive Anisotropic Textured Gaussians for Efficient 3D Scene Representation

**Sheng-Chi Hsu**     **Ting-Yu Yen**     **Shih-Hsuan Hung**     **Hung-Kuo Chu**

National Tsing Hua University

## Abstract

Gaussian Splatting has emerged as a powerful representation for high-quality, real-time 3D scene rendering. While recent works extend Gaussians with learnable textures to enrich visual appearance, existing approaches allocate a fixed square texture per primitive, leading to inefficient memory usage and limited adaptability to scene variability. In this paper, we introduce **adaptive anisotropic textured Gaussians** (A²TG), a novel representation that generalizes textured Gaussians by equipping each primitive with an anisotropic texture. Our method employs a gradient-guided adaptive rule to jointly determine texture resolution and aspect ratio, enabling non-uniform, detail-aware allocation that aligns with the anisotropic nature of Gaussian splats. This design significantly improves texture efficiency, reducing memory consumption while enhancing image quality. Experiments on multiple benchmark datasets demonstrate that A²TG consistently outperforms fixed-texture Gaussian Splatting methods, achieving comparable rendering fidelity with substantially lower memory requirements.

## 1 Introduction

Recent advances in Gaussian Splatting have established it as a powerful paradigm for 3D scene representation, offering both high-fidelity reconstructions and real-time rendering performance. In particular, 3D Gaussian Splatting (3DGS) and its 2D variant, 2D Gaussian Splatting (2DGS), have attracted significant attention due to their ability to combine explicit geometric structure with efficient differentiable rasterization, enabling compelling results across a range of applications in view synthesis, reconstruction, and immersive content creation.

Unlike mesh-based representations, which compactly encode repetitive high-frequency appearance via texture maps, Gaussian Splatting typically approximates such detail by instantiating many small primitives, leading to artifacts and inefficient memory usage. Chao et al. (2025) show that attaching a learnable, uniform square texture patch to each Gaussian substantially enhances visual richness and realism. However, this design ignores variability in primitive geometry and opacity. Some Gaussians require high-frequency textures to capture fine detail, whereas others only need to convey coarse appearance with simple texture. Consequently, fixed-size texture allocations waste memory, inflate storage, and poorly adapt to the anisotropic Gaussians.

In this paper, we propose **adaptive anisotropic textured Gaussians** (A²TG), a novel representation designed to address these limitations. Instead of attaching a uniform square texture to each Gaussian, A²TG assigns an anisotropic texture whose resolution and aspect ratio are jointly determined by a gradient-based adaptive texture control strategy. This adaptive anisotropic texture allows Gaussians to have different patterns and details across the scene. Gaussians covering high-frequency or directionally elongated regions are assigned higher-resolution anisotropic textures, while simpler regions receive more compact representations. However, it is challenging to devise a principle to guide per-Gaussian texture resolution and pixel values. Many 2D Gaussians yield little benefit from textures due to small projected footprints, low opacity, and occlusion. We introduce **gradient-guided adaptive control** to iteratively upscale the textures according to the positional gradient and the geometry

of the Gaussians. As a result, A$^2$TG improves memory efficiency while also achieving comparable image quality, as texture parameters are more efficiently allocated.

We evaluate A$^2$TG across diverse datasets and benchmark tasks. Our results show that the proposed approach achieves more effective texture utilization, leading to comparable image quality compared to textured Gaussian Splatting methods with fixed square textures, while substantially reducing memory overhead. These findings suggest that our adaptive anisotropic texture allocation is a promising path toward scalable and efficient textured Gaussian representations.

Our main contributions are summarized as follows.

- We introduce A$^2$TG, a generalization of Textured Gaussians that leverages an adaptive anisotropic texture for each Gaussian.

- We propose a gradient-based adaptive texture control strategy for allocating texture resolution and aspect ratio, enabling non-uniform, detail-aware texture mapping.

- We demonstrate that A$^2$TG improves memory efficiency while maintaining comparable visual fidelity across multiple benchmarks, advancing the state-of-the-art in texture-based Gaussian Splatting.

## 2 RELATED WORK

### 2.1 NOVEL VIEW SYNTHESIS

Novel View Synthesis addresses the challenge of generating realistic renderings of 3D scenes from previously unseen viewpoints, given a set of captured images with known camera poses (Chaurasia et al., 2013; Hedman et al., 2018b; Kopanas et al., 2021). Neural Radiance Fields (NeRF) represent scenes using a multi-layer perceptron (MLP) that models both geometry and view-dependent appearance, trained through volume rendering to achieve high-fidelity image synthesis (Barron et al., 2021; Tancik et al., 2020; Mildenhall et al., 2021). On the other hand, 3D Gaussian Splatting (3DGS) (Kerbl et al., 2023) has emerged as a powerful alternative, capable of delivering real-time novel view synthesis with impressive visual quality. Building on its success, extensions of 3DGS have rapidly appeared across diverse applications. More recently, 2D Gaussian Splatting (2DGS) (Huang et al., 2024) builds upon 3DGS by introducing a "flattened" variant of Gaussian primitives that aligns more naturally with object surfaces, allowing more accurate intersection calculation and opening the door to further applications such as mesh extraction.

### 2.2 TEXTURED GAUSSIAN SPLATTING REPRESENTATION

The color of each Gaussian primitive is typically parameterized by spherical-harmonic (SH) coefficients, which, given a viewing direction, evaluate to a view-dependent RGB. Consequently, for a fixed view, each primitive yields a single color distributing over its 2D footprint, limiting intra-primitive color detail. Huang & Gong (2024) adds spatially varying appearance to each 3D Gaussian, enabling richer 3DGS representation. Texture-GS (Xu et al., 2024b) maps textures onto 3D Gaussians, but the method is demonstrated primarily on small objects, limiting scalability. Chao et al. (2025) further introduces RGBA texture maps to improve visual quality.

By contrast, 2DGS (Huang et al., 2024) conforms more closely to underlying surfaces and provides a more accurate parameterization for texture mapping, making it better suited for texture integration (Svitov et al., 2024; Xu et al., 2024a; Weiss & Bradley, 2024; Rong et al., 2025; Song et al., 2024). These methods employ predefined RGB or RGBA textures, and (Rong et al., 2025) further adopts world-space texture mapping. However, these methods assign a fixed-size and square texture to every 2D Gaussian. Because Gaussians have varying geometry and opacity, uniform texture sizes are memory-inefficient; tiny or highly transparent Gaussians gain little benefit while inflating storage. Thus, we propose A$^2$TG, which uses adaptive anisotropic textured Gaussians that allocate texture resolution to each primitive according to its footprint and gradient.

### 2.3 Memory-Efficient Gaussian Splatting Representation

Gaussian primitives offer efficient real-time rendering via learned positions, covariances, color, and opacity; however, 3DGS models are typically storage-intensive due to the large number of primitives and per-primitive attributes. Several 3DGS compression methods address this issue, including vector quantization (Fan et al., 2024; Lee et al., 2024; Niedermayr et al., 2024), and splat-count reduction (Papantonakis et al., 2024; Zhang et al., 2024; Mallick et al., 2024), and context models (Wang et al., 2024) . These approaches can improve memory efficiency, while the results have similar visual fidelity to the original Gaussian models (Bagdasarian et al., 2025). In this work, we target memory-efficient textured Gaussians by adaptively controlling per-Gaussian texture resolution; this direction is orthogonal and complementary to prior compression methods and remains compatible with them.

## 3 Preliminaries

Textured Gaussian splatting methods naturally build upon **2D Gaussian splatting** (Huang et al., 2024), as the latter provides a convenient framework for defining $uv$-coordinates in the local space of 2D splats, thereby enabling consistent texture sampling.

More concretely, 2D Gaussian Splatting represents a 3D scene using a collection of flat 2D Gaussians instead of volumetric 3D Gaussians (Kerbl et al., 2023). Each Gaussian is parameterized as

$$\{\boldsymbol{\mu}_i, \mathbf{s}_i, \mathbf{r}_i, o_i, \boldsymbol{c}_i^{\mathrm{SH}}\},$$

where $\boldsymbol{\mu}_i$ denotes its center position, $\mathbf{s}_i$ the 2D scale, $\boldsymbol{r}_i$ a quaternion rotation, $o_i$ the opacity, and $\boldsymbol{c}_i^{\mathrm{SH}}$ the spherical harmonics coefficients encoding view-dependent appearance.

For each 2D Gaussian, a local $uv$-coordinates system is defined. The mapping from this local space to the screen space is expressed as

$$\mathbf{x} = \mathbf{WH}(u, v, 1, 1)^{\top}, \tag{1}$$

where $\mathbf{x} = (x, y, 1, 1)^{\top}$ is the homogeneous ray passing through pixel $(x, y)$, $(u, v, 1, 1)^{\top}$ is the corresponding intersection point in the local Gaussian space, $\mathbf{W} \in \mathbb{R}^{4 \times 4}$ is the world-to-screen transformation matrix, and $\mathbf{H}$ is the local-to-world transformation constructed from the Gaussian's position, rotation, and scale (Huang et al., 2024).

To render a set of Gaussians, $uv$-coordinates of the ray–splat intersection are first computed as $\mathbf{u}(\mathbf{x}) = (u(\mathbf{x}), v(\mathbf{x}))$ by inverting Equation 1. The final color of each pixel is then obtained using front-to-back alpha compositing:

$$\mathbf{c}(\mathbf{x}) = \sum_{i=1} \mathbf{c}_i \, o_i \, \mathcal{G}_i(\mathbf{u}(\mathbf{x})) \prod_{j=1}^{i-1} \left(1 - o_j \, \mathcal{G}_j(\mathbf{u}(\mathbf{x}))\right), \tag{2}$$

where $\mathbf{c}_i$ is the color of the $i$-th Gaussian and $\mathcal{G}_i$ is its Gaussian kernel evaluated at the $uv$-coordinates.

## 4 Methodology

Recent works augment 2D Gaussians with texture maps to improve rendering quality and efficiency. Most assign a fixed, square texture to every Gaussian, regardless of its scale, opacity, or visibility. This creates redundancy and unnecessary memory usage. We instead assign each 2D Gaussian an individual texture resolution and shape, thereby allocating parameters where they matter most. Our framework starts with training the given scene by integrating a Markov Chain Monte Carlo (MCMC) densification scheme (Kheradmand et al., 2024) into the 2DGS pipeline (Huang et al., 2024) for the first $30,000$ iterations. With MCMC densification, we can easily control the total number of Gaussians in the final results. During this pretraining stage, each Gaussian's texture is fixed to $1 \times 1$ with texture color $T^{\mathrm{RGB}} = 0$ and texture alpha $T^A = 0$ (i.e., without textures). Next, we fix the total number of 2D Gaussians, set the texture alpha $T^A = 1$, and train the 2DGS model with the gradient-based adaptive texture control for another $30,000$ iterations. At each iteration, we

| (a) Optimization | (b) Gradient-Driven Gaussian Selection | (c) Adaptive Texture Upscaling |

Figure 1: **Overview of gradient-based adaptive texture control.** Given an initial 2DGS model, (a) our system first optimizes the parameters of the 2D Gaussians and their textures. (b) Next, we compute the positional gradient of the textured 2D Gaussians (as depicted in gray blocks) and select the 2D Gaussians that need to increase the resolution of the texture to capture more details. (c) Finally, we adaptively upscale the textures according to the anisotropy of the Gaussians.

first update the parameters of the 2D Gaussians and their textures (i.e., $T^{\mathrm{RGB}}$ and $T^{\mathrm{A}}$), followed by applying *gradient-based adaptive texture control*, which consists of *gradient-driven Gaussian selection* and *adaptive texture upscaling* as illustrated in Figure 1. This procedure adjusts the resolution and aspect ratio of each texture to match the anisotropic footprint of its Gaussian, upscaling where gradients are large. We detail the gradient-based adaptive texture control in the following subsection.

## 4.1 ANISOTROPIC TEXTURED GAUSSIANS

We first augment each 2D Gaussian an RGB texture $T_i^{\mathrm{RGB}}$ and an alpha texture $T_i^{\mathrm{A}}$, then we utilize the local space of the 2D Gaussians for texture mapping with RGBA textures, as the prior works of textured Gaussians (Xu et al., 2024a; Svitov et al., 2024; Chao et al., 2025). After getting the $uv$-coordinates by taking the inverse of Equation 1, we rescale the range of $u$ and $v$ from $[-1, 1]$ to $[0, \mathcal{T}_i^u]$ and $[0, \mathcal{T}_i^v]$, where $\mathcal{T}_i^u$ and $\mathcal{T}_i^v$ are the texture width and height of the $i$-th 2D Gaussian, so every texture may have different width and height.

The color contribution of the $i$-th 2D Gaussian to pixel $\mathbf{x}$ is now calculated by combining the RGB color from the spherical harmonic $\mathbf{c}_i^{\mathrm{SH}}$ and the texture color as

$$c_i(\mathbf{x}) = \mathbf{c}_i^{\mathrm{SH}} + T_i^{\mathrm{RGB}}\left(\mathbf{u}(\mathbf{x})\right), \tag{3}$$

where $\mathbf{u}$ maps the pixel $\mathbf{x}$ to the $uv$-coordinates. The alpha value of the $i$-th Gaussian is calculated by

$$\alpha_i(\mathbf{x}) = o_i \cdot \mathcal{G}(\mathbf{u}(\mathbf{x})) \cdot T_i^{\mathrm{A}}\left(\mathbf{u}(\mathbf{x})\right), \tag{4}$$

where $\mathcal{G}$ and $o_i$ are the Gaussian distribution function and the opacity of the $i$-th 2D Gaussian. For querying the texture values $T_i^{\mathrm{RGB}}$ and $T_i^{\mathrm{A}}$ from $uv$-coordinates $\mathbf{u}(\mathbf{x})$, we use bilinear interpolation, following (Chao et al., 2025). Finally, the $k$-th channel of the resulting pixel value at pixel $\mathbf{x}$ is calculated on the sorted 2D Gaussians from front to back as

$$\mathbf{c}^k(\mathbf{x}) = \sum_{i=1} c_i^k(\mathbf{x})\alpha_i(\mathbf{x}) \prod_{j=1}^{i-1}(1 - \alpha_j(\mathbf{x})), \tag{5}$$

where $c_i^k$ is the $k$-th channel color of the $i$-th Gaussian. We next utilize the pixel and alpha values to compute the gradient and guide the upscaling of the textures.

## 4.2 GRADIENT-DRIVEN GAUSSIAN SELECTION

It's difficult to design a principle to determine the needs of resolution and pixel values of textures for each 2D Gaussian. First, some 2D Gaussians gain little from textures because their projected footprint covers only a few pixels in the training views or their opacity is very low. Second, occlusion by foreground splats limits the utility of textures even when a Gaussian's opacity is high. Finally, in regions with homogeneous appearance (e.g., uniformly colored walls), textures provide minimal benefit regardless of Gaussian size or opacity. Therefore, we introduce the gradient-driven Gaussian selection, similar to the densification procedure from 3DGS (Kerbl et al., 2023; Ye et al., 2024).

Given a rendered view of the 2DGS and its corresponding training view, let $\mathcal{L}$ be the $L_1$ and SSIM loss between the views at pixel $\mathbf{x}$ and $\boldsymbol{\mu}_i = (\mu_{i,x}, \mu_{i,y}, \mu_{i,z})^\top$ be the position of the $i$-th Gaussian. The positional gradient from the pixel at coordinate $\mathbf{x}$ with respect to the $i$-th Gaussian is calculated as $\nabla_{\boldsymbol{\mu}_i}\mathcal{L} = \left( \frac{\partial \mathcal{L}}{\partial \mu_{i,x}}, \frac{\partial \mathcal{L}}{\partial \mu_{i,y}}, \frac{\partial \mathcal{L}}{\partial \mu_{i,z}} \right)^\top$. We then derive the $\frac{\partial \mathcal{L}}{\partial \mu_{i,x}}$ with the pixel value and the alpha value of the $i$-th Gaussian as

$$\frac{\partial \mathcal{L}}{\partial \mu_{i,x}} = \sum_{k=1}^{3} \frac{\partial \mathcal{L}}{\partial \mathbf{c}^k} \cdot \frac{\partial \mathbf{c}^k}{\partial \alpha_i} \cdot \frac{\partial \alpha_i}{\partial \mu_{i,x}}. \tag{6}$$

The first term corresponds to our idea of using pixel differences to guide texture upscaling, which eliminates the possibility of introducing additional parameters on already well-reconstructed regions of the scene. The last two terms involved in the positional gradient calculation contain the information of occlusion from other Gaussians, the opacity, and the pixel contribution count. They can be computed as

$$\frac{\partial \mathbf{c}^k}{\partial \alpha_i} = \prod_{l=1}^{i-1}(1 - \alpha_l)c_i^k + \sum_{p=i+1}^{N} c_p^k \frac{\partial w_p}{\partial \alpha_i}, \tag{7}$$

where $\frac{\partial w_p}{\partial \alpha_i} = -\alpha_p \prod_{\substack{l=1 \\ l \neq i}}^{p-1}(1 - \alpha_l)$, and

$$\frac{\partial \alpha_i}{\partial \mu_{i,x}} = o_i \cdot \frac{\partial \mathcal{G}_i(\mathbf{u})}{\partial \mu_{i,x}} \cdot T_i^{\mathrm{A}}(\mathbf{u}). \tag{8}$$

Similarly, we can derive the equations for $\frac{\partial \mathcal{L}}{\partial \mu_{i,y}}$ and $\frac{\partial \mathcal{L}}{\partial \mu_{i,z}}$. For the $i$-th Gaussian, we then accumulate the absolute value of its positional gradient over the covered pixels as $\nabla_{\boldsymbol{\mu}_i}\mathcal{L}$, following AbsGS(Ye et al., 2024). If the magnitude of the accumulated gradient exceeds a threshold (i.e., $\|\nabla_{\boldsymbol{\mu}_i}\mathcal{L}\|_2 > k_{\mathrm{G}}$), it indicates the presence of high-frequency content on the Gaussian. We then select the Gaussian as a candidate for adaptive texture upscaling.

### 4.3 Adaptive Texture Upscaling

After selecting candidate 2D Gaussians, we adaptively increase their texture resolution based on anisotropy. Specifically, we compute the ratio of the two semi-axes, $\mathrm{s}_x$ and $\mathrm{s}_y$, of each 2D Gaussian and use it to determine the anisotropy. The texture resolutions $\mathcal{T}^u$ and $\mathcal{T}^v$ are then updated according to the following rules:

- If $\mathrm{s}_x/\mathrm{s}_y > k_{\mathrm{A}}$ and $s_y < k_{\mathrm{S}}$, then we double $\mathcal{T}^u$.
- If $\mathrm{s}_y/\mathrm{s}_x > k_{\mathrm{A}}$ and $s_x < k_{\mathrm{S}}$, then we double $\mathcal{T}^v$.
- Otherwise, we double the resolution of both $\mathcal{T}^u$ and $\mathcal{T}^v$.

Here, $k_{\mathrm{A}}$ and $k_{\mathrm{S}}$ are predefined thresholds. This adaptive strategy allocates resolution more effectively: textures are refined preferentially along the dimension most sensitive to gradient variations, while still allowing isotropic refinement when Gaussians are approximately square or both axes require higher detail.

When upscaling, the new texture color $T^{\mathrm{RGB}}$ and texture alpha $T^A$ are initialized from the nearest pixel of the previous texture. Both $T^{\mathrm{RGB}}$ and $T^A$, together with the Gaussian parameters, are then optimized in subsequent iterations. The adaptive texture upscaling is applied every 500 iterations, allowing gradients to accumulate before resolution adjustments.

## 5 Experiments

### 5.1 Experimental Setup

**Datasets and metrics.** We conducted experiments on the Mip-NeRF 360 dataset (Barron et al., 2022)(7 scenes), the Tanks and Temples dataset (Knapitsch et al., 2017)(2 scenes), and the Deep Blending dataset (Hedman et al., 2018a)(2 scenes). The evaluation metrics include PSNR, SSIM,

| Method | Mip-NeRF 360 | | | | | Tanks and Temples | | | | | DeepBlending | | | | |
|---|---|---|---|---|---|---|---|---|---|---|---|---|---|---|---|
| | PSNR ↑ | SSIM ↑ | LPIPS ↓ | #GS | Mem | PSNR ↑ | SSIM ↑ | LPIPS ↓ | #GS | Mem | PSNR ↑ | SSIM ↑ | LPIPS ↓ | #GS | Mem |
| 2DGS*-MCMC | 28.30 | 0.836 | 0.177 | 862k | 200.00MB | 23.16 | 0.827 | 0.151 | 862k | 200.00MB | 29.63 | 0.898 | 0.192 | 862k | 200.00MB |
| Super Gaussians | 28.31 | 0.839 | 0.232 | 675k | 199.75MB | 23.49 | 0.841 | 0.195 | 674k | **199.49MB** | 29.52 | 0.905 | 0.260 | 675k | 199.72MB |
| BBSplat | 26.72 | 0.792 | 0.257 | 46k | 200.00MB | 22.51 | 0.794 | 0.241 | 46k | 200.00MB | 28.73 | 0.893 | 0.275 | 46k | 200.00MB |
| Textured Gaussians* | 28.37 | 0.832 | 0.188 | 410k | 200.00MB | 23.41 | 0.824 | 0.164 | 410k | 200.00MB | 29.51 | 0.897 | 0.198 | 410k | 200.00MB |
| A²TG (Ours) | 28.51 | 0.838 | 0.174 | 700k | 199.68MB | 23.56 | 0.828 | 0.153 | 690k | 199.51MB | 29.86 | 0.900 | 0.187 | 700k | **189.42MB** |
| 2DGS* | 26.72 | 0.787 | 0.308 | 259k | 60.00MB | 22.32 | 0.791 | 0.275 | 216k | 50.00MB | 28.89 | 0.889 | 0.302 | 259k | 60.00MB |
| 2DGS*-MCMC | 27.46 | 0.806 | 0.228 | 259k | 60.00MB | 22.58 | 0.803 | 0.199 | 216k | 50.00MB | 29.24 | 0.893 | 0.217 | 259k | 60.00MB |
| Super Gaussians | 25.29 | 0.791 | 0.300 | 202k | 59.93MB | 21.46 | 0.783 | 0.273 | 169k | 49.92MB | 29.01 | 0.895 | 0.292 | 203k | 59.96MB |
| BBSplat | 23.07 | 0.614 | 0.462 | 14k | 60.00MB | 16.21 | 0.522 | 0.579 | 12k | 50.00MB | 26.19 | 0.839 | 0.385 | 14k | 60.00MB |
| Textured Gaussians* | 27.33 | 0.795 | 0.253 | 123k | 60.00MB | 22.60 | 0.789 | 0.231 | 102k | 50.00MB | 29.19 | 0.892 | 0.229 | 123k | 60.00MB |
| A²TG (Ours) | 27.47 | 0.805 | 0.233 | 207k | **58.02MB** | 22.73 | 0.798 | 0.213 | 160k | **49.09MB** | 29.42 | 0.895 | 0.215 | 200k | **59.55MB** |

Table 1: **Quantitative comparison under a fixed memory budget.** We report PSNR ↑, SSIM ↑, LPIPS ↓, number of Gaussians (#GS), and memory (Mem, MB). The top three results are highlighted in red , orange , and yellow , and the least memory sizes are in **bold** .

LPIPS, the number of Gaussians, and the memory size of trainable parameters for each algorithm. The testing set was constructed by selecting every eighth image, while the remaining images were used for training.

**Baselines.** We compare A²TG against 2DGS (Huang et al., 2024), and textured Gaussian methods including Textured Gaussians (Chao et al., 2025), BBSplat (Svitov et al., 2024), and SuperGaussians (Xu et al., 2024a).

To ensure a fair comparison, we disable the depth-distortion and normal-consistency losses in 2DGS, as these terms primarily improve mesh quality at the cost of slightly reduced image fidelity. Furthermore, since both Textured Gaussians and A²TG are built on 2DGS with an MCMC-based density control, we report results for two 2DGS variants: the original 2DGS with the adaptive density control of Kerbl et al. (2023) and 2DGS equipped with the same MCMC strategy(Kheradmand et al., 2024).

We use an unofficial implementation of Textured Gaussians that employs 2DGS for rasterization instead of 3DGS. We denote the modified baselines as 2DGS*, 2DGS*-MCMC, and Textured Gaussians*, respectively.

**Implementation details.** Textured Gaussians* also adopts a two-stage training process, beginning with an MCMC pretraining. For fairness, all experiments with Textured Gaussians* and our A²TG model share the same first-stage results when trained with the same number of Gaussians. In the second stage, Textured Gaussians use a fixed texture resolution of $4 \times 4$, while our A²TG method applies adaptive texture upscaling at iterations 500 and 1000. As a result, final textures in Textured Gaussians remain $4 \times 4$, whereas those in A²TG vary within $\{1, 2, 4\} \times \{1, 2, 4\}$, including both square and anisotropic shapes. The parameters of A²TG are set as $k_A = 4.0$, $k_S = 0.01$, $k_G = 0.00002$.

## 5.2 COMPARISONS

We employ two different comparison settings for a comprehensive evaluation on the performance of our method, one with fixed memory budget, and the other with fixed number of Gaussians. Metrics including PSNR, SSIM, LPIPS on all the datasets are reported separately. Note that BBSplat is designed for a lower number of Gaussians because of their extra parameters; thus, they were not evaluated under the setting of 500k and 1M Gaussians.

**Comparison under a fixed memory budget.** We report the visual quality metrics, number of Gaussians, and memory consumption of trainable parameters of all the baselines and our A²TG method in Table 1. Our method has achieved the best performance across various textured-based 2DGS works under the same memory constraint, demonstrating the efficiency of our method. Since our adaptive texture control is based on gradients threshold selection, we cannot achieve the exact same memory consumption as other methods, but our A²TG method still achieves higher overall visual quality with less memory especially for PSNR. Other texture-based methods suffer from excessive texture parameters, thus resulting in worse visual quality under a fixed memory constraint.

| Method | Mip-NeRF 360 | | | | Tanks & Temples | | | | DeepBlending | | | |
|---|---|---|---|---|---|---|---|---|---|---|---|---|
| | PSNR ↑ | SSIM ↑ | LPIPS ↓ | Mem(MB,+%) | PSNR ↑ | SSIM ↑ | LPIPS ↓ | Mem(MB,+%) | PSNR ↑ | SSIM ↑ | LPIPS ↓ | Mem(MB,+%) |
| *#GS = 1M* | | | | | | | | | | | | |
| 2DGS* | – | – | – | – | – | – | – | – | 29.58 | 0.899 | 0.263 | 232 (0%) |
| 2DGS*-MCMC | 28.37 | 0.838 | 0.172 | 232 (0%) | 23.11 | 0.827 | 0.148 | 232 (0%) | 29.60 | 0.898 | 0.190 | 232 (0%) |
| Super Gaussians | 28.39 | 0.848 | 0.218 | 296 (28%) | 23.66 | 0.847 | 0.185 | 284 (**22%**) | 29.38 | 0.905 | 0.253 | 295 (27%) |
| Textured Gaussians* | 28.81 | 0.846 | 0.159 | 488 (110%) | 23.58 | 0.833 | 0.141 | 488 (110%) | 29.64 | 0.897 | 0.185 | 488 (110%) |
| A²TG (Ours) | 28.70 | 0.844 | 0.163 | 291 (**25%**) | 23.58 | 0.831 | 0.145 | 292 (26%) | 29.82 | 0.900 | 0.180 | 277 (**19%**) |
| *#GS = 500k* | | | | | | | | | | | | |
| 2DGS* | 27.77 | 0.811 | 0.270 | 116 (0%) | 23.08 | 0.824 | 0.220 | 116 (0%) | 29.45 | 0.894 | 0.279 | 116 (0%) |
| 2DGS*-MCMC | 27.95 | 0.824 | 0.196 | 116 (0%) | 22.87 | 0.819 | 0.166 | 116 (0%) | 29.45 | 0.896 | 0.202 | 116 (0%) |
| Super Gaussians | 28.01 | 0.831 | 0.245 | 148 (28%) | 23.31 | 0.835 | 0.206 | 148 (28%) | 29.49 | 0.903 | 0.266 | 148 (28%) |
| Textured Gaussians* | 28.47 | 0.836 | 0.181 | 244 (110%) | 23.50 | 0.826 | 0.158 | 244 (110%) | 29.60 | 0.898 | 0.195 | 244 (110%) |
| A²TG (Ours) | 28.31 | 0.832 | 0.187 | 148 (**28%**) | 23.43 | 0.824 | 0.163 | 150 (29%) | 29.71 | 0.899 | 0.193 | 140 (**21%**) |
| *#GS = 100k* | | | | | | | | | | | | |
| 2DGS* | – | – | – | – | 19.17 | 0.734 | 0.324 | 23 (0%) | 28.32 | 0.879 | 0.324 | 23 (0%) |
| 2DGS*-MCMC | 26.40 | 0.767 | 0.294 | 23 (0%) | 22.07 | 0.776 | 0.245 | 23 (0%) | 28.69 | 0.883 | 0.253 | 23 (0%) |
| Super Gaussians | 23.12 | 0.737 | 0.366 | 30 (**28%**) | 20.11 | 0.737 | 0.327 | 30 (**28%**) | 26.61 | 0.875 | 0.331 | 30 (28%) |
| BBSplat | 27.73 | 0.828 | 0.216 | 432 (1764%) | 23.40 | 0.842 | 0.164 | 432 (1764%) | 28.78 | 0.894 | 0.268 | 432 (1764%) |
| Textured Gaussians* | 27.04 | 0.785 | 0.268 | 49 (110%) | 22.52 | 0.787 | 0.233 | 49 (110%) | 29.18 | 0.890 | 0.236 | 49 (110%) |
| A²TG (Ours) | 26.62 | 0.775 | 0.283 | 30 (**28%**) | 22.37 | 0.780 | 0.243 | 31 (32%) | 29.04 | 0.888 | 0.243 | 29 (**24%**) |

Table 2: **Quantitative comparison under a fixed number of Gaussians.** We report PSNR ↑, SSIM ↑, LPIPS ↓, and memory (MB). The parameters increase for texture parameters relative to the size of 2DGS is added as percentage after memory (MB). The top three results are highlighted in red, orange, and yellow, and the least parameter increases are in **bold**.

For visual comparison, we present the testing-view results of A²TG alongside 2DGS* and Textured Gaussians* in Figure 3. Owing to the flexibility of textured representations, both Textured Gaussians* and A²TG achieve notably higher visual quality than 2DGS*, despite the latter using a larger number of Gaussians. Moreover, under equal memory budgets, A²TG produces higher-fidelity renderings than Textured Gaussians*, primarily because it can represent the scene with a larger set of Gaussians.

**Comparison under a fixed number of Gaussians.** We report the visual quality metrics and the memory consumption in Table 2. The percentage increase in memory is measured relative to 2DGS* with the number of Gaussians fixed to 1M, 500k and 100k. All methods outperform 2DGS due to the textures. BBSplat and Textured Gaussians* typically have slightly better visual quality than our method but require substantially more memory because of the inefficient ways to use the texture parameters. In particular, Textured Gaussians* takes up to more than four times texture memory than A²TG with around $0.4$ dB increase in PSNR. We also report the full experiments in Appendix A.1.

To better illustrate the efficiency of our method, we compare our A²TG, Textured Gaussians and 2DGS*-MCMC for different numbers of Gaussians, and plot the relation between memory and PSNR, as well as the relation between point count and memory as shown in Figure 2. Overall, our method not only has a better reconstruction quality under same memory constraint, but also with less memory usage than Textured Gaussians under the same number of Gaussians.

**Distribution of upscaled textures.** To illustrate A²TG's efficiency on a complex scene, we color-code Gaussians: blue for $2 \times 2$ or $4 \times 4$ textures, red for non-square textures, and the original color for $1 \times 1$ (no upscaling). Figure 4 (right) shows the Garden scene from the Mip-NeRF 360 dataset, while Figure 4 (left) plots the percentage distribution of all the texture resolutions. Notably, $62.4\%$ of Gaussians retain $1 \times 1$ textures under our gradient-driven Gaussian selection, indicating that texture resolution is allocated sparingly where additional detail is unnecessary. Red highlights concentrate along sharp edges (e.g., gaps between table planks), showing that A²TG detects thin anisotropic Gaussians and assigns non-square textures accordingly.

**Texture Decomposition.** To better understand the role of adaptive textures in A²TG, we perform a qualitative ablation by selectively disabling components of the appearance model. Figure 5 shows three versions of each scene: (1) the full A²TG rendering, (2) a version without textures, and (3) a version without base color. For the "no textures" variant, we set all RGB textures $T^{\text{RGB}}$ to zero and all alpha textures $T^{\text{A}}$ to one, removing texture-driven modulation. For the "no base color" variant, we set the spherical harmonics color $\mathbf{c}_i^{\text{SH}}$ to zero, leaving appearance determined solely by the learned textures.

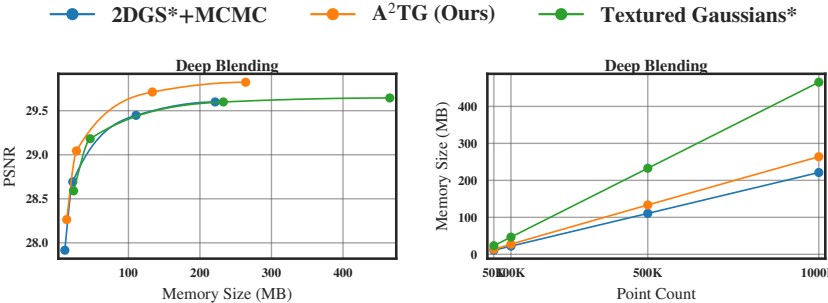

Figure 2: **Comparison of 2DGS, A²TG and Textured Gaussians on the DeepBlending datasets.** Left: PSNR versus memory size (MB). Right: memory size (MB) versus point count. A²TG achieves higher reconstruction quality under the same memory budget and requires less memory than Textured Gaussians for the same number of Gaussians.

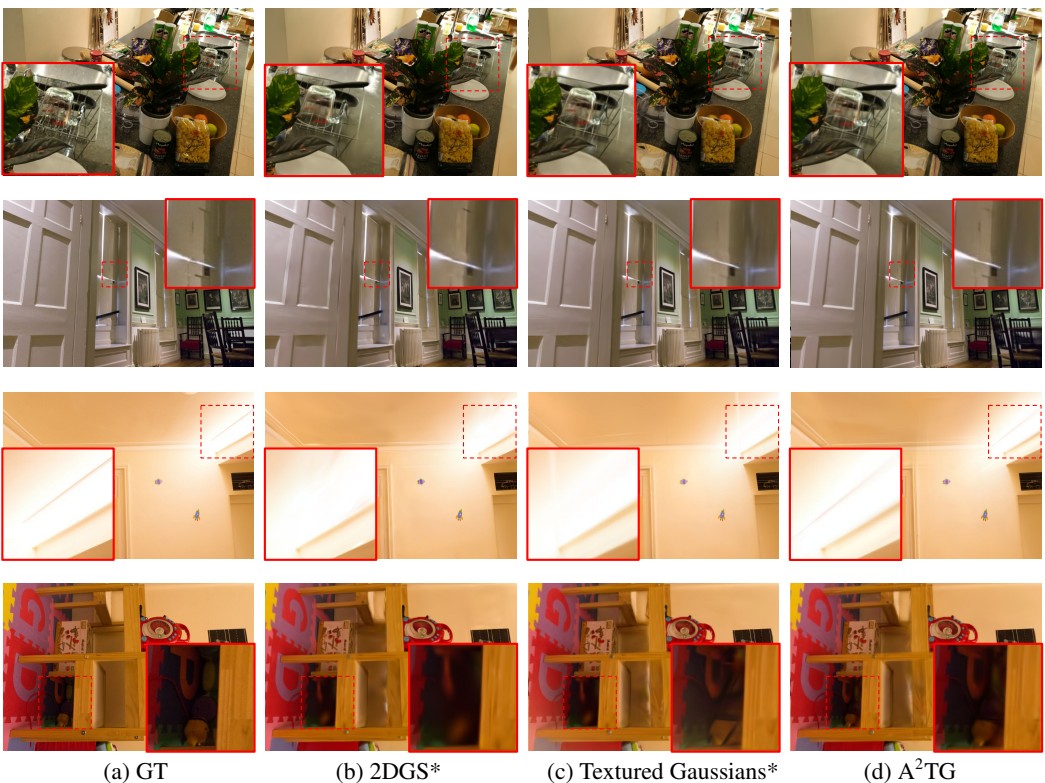

Figure 3: **Qualitative comparisons.** We show the qualitative comparisons of 2DGS* and Textured Gaussians* with A²TG under fixed memory constraint from the Mip-NeRF360 datasets and the DeepBlending datasets. With textures, both Textured Gaussians* and A²TG reconstruct fine scene details, whereas A²TG uses less memory.

As shown in Figure 5, removing textures leads to a clear loss of high-frequency detail—e.g., leaf boundaries, grass structure, and fine variations in carpets and fabrics—indicating that adaptive textures capture view-consistent residual appearance beyond what low-order spherical harmonics can express. Since every Gaussian contains at least $1 \times 1$ texture, removing textures affects all regions.

Conversely, removing the SH base color produces overly dark renderings with missing global shading. Although textures recover much of the fine detail, the absence of SH color removes low-frequency illumination and structural coherence, showing that the SH term provides a smooth, lighting-aware foundation.

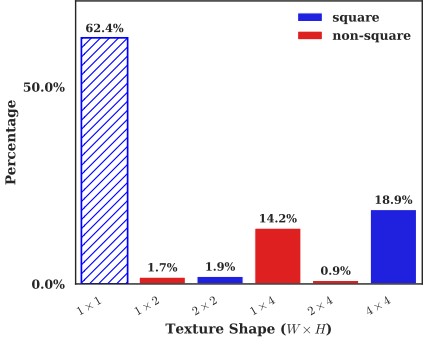

(a) Texture Resolution Percentage

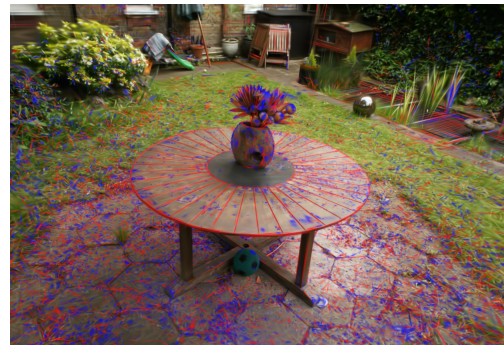

(b) Texture Resolution Distribution

Figure 4: **Texture resolution percentage and distribution.** This figure shows the percentage and distribution of texture resolution produced by the adaptive texture upscaling on the scene *garden* from Mip-Nerf360 dataset. Gaussians highlighted in blue have square texture of $2 \times 2$ and $4 \times 4$, and those highlighted in red have non-square texture resolution.

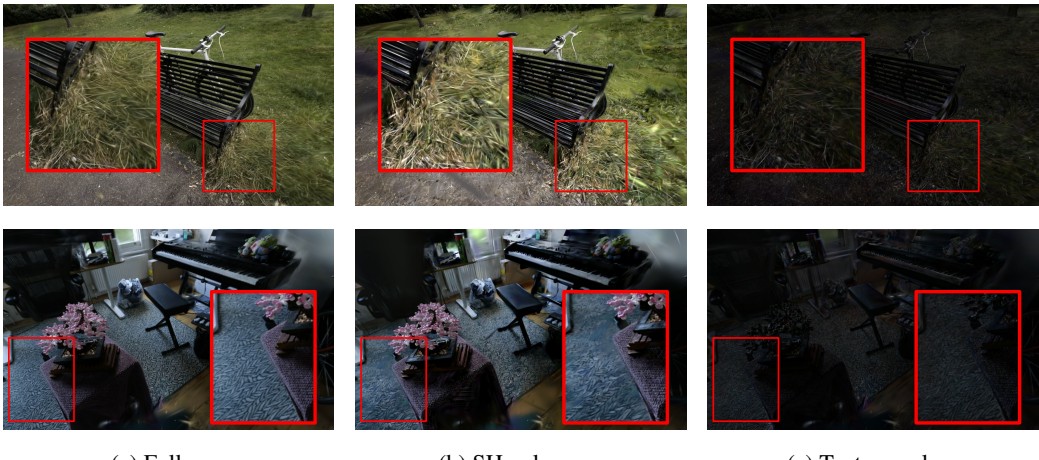

| (a) Full | (b) SH only | (c) Texture only |

Figure 5: **Qualitative visualization of what the adaptive textures learn. Left:** full rendering from $A^2TG$. **Middle:** rendering without textures (RGB textures set to zero, alpha textures set to one). **Right:** rendering without SH base color. The comparison, visualized on two scenes shows that textures capture high-frequency residual appearance such as foliage structure and fabric detail, while SH color provides smooth, low-frequency shading. Together, they produce the final photorealistic result.

Overall, these ablations highlight the complementary roles of SH color and adaptive textures: SH models coarse shading and global appearance, while textures encode local, high-frequency details. This further supports our choice of allocating texture resolution adaptively so that texture capacity is focused where fine detail is needed.

## 5.3 ABLATION STUDIES

We examine the impact of resolution scaling and anisotropy on our adaptive texture upscaling. Ablations are run on three datasets with the number of Gaussians fixed at 50k, 100k, 500k and 1M, comparing $A^2TG$ to two variants: (i) without texture upscaling (w/o Upscaling) and (ii) without anisotropic texture upscaling (w/o Anisotropy). Table 3 reports average memory usage and reconstruction quality (PSNR, SSIM, LPIPS). Removing scaling yields the lowest memory but the worst quality. Disabling anisotropy produces quality close to the full method but uses more memory because square textures cannot match anisotropic Gaussians as efficiently.

| Method | PSNR↑ | SSIM↑ | LPIPS↓ | Mem↓ | Method | PSNR↑ | SSIM↑ | LPIPS↓ | Mem↓ |
|---|---|---|---|---|---|---|---|---|---|
| *#GS = 1M* | | | | | *#GS = 100k* | | | | |
| w/o Upscaling | 27.03 | 0.854 | 0.170 | 232.00 | w/o Upscaling | 25.72 | 0.809 | 0.264 | 23.20 |
| w/o Anisotropy | 27.38 | 0.859 | 0.162 | 298.22 | w/o Anisotropy | 26.01 | 0.815 | 0.257 | 31.10 |
| Ours | 27.37 | 0.858 | 0.163 | 286.54 | Ours | 26.01 | 0.814 | 0.256 | 29.67 |
| *#GS = 500k* | | | | | *#GS = 50k* | | | | |
| w/o Upscaling | 26.76 | 0.846 | 0.188 | 116.00 | w/o Upscaling | 24.87 | 0.780 | 0.318 | 11.60 |
| w/o Anisotropy | 27.19 | 0.852 | 0.181 | 152.10 | w/o Anisotropy | 25.19 | 0.787 | 0.308 | 15.51 |
| Ours | 27.15 | 0.852 | 0.181 | 145.92 | Ours | 25.17 | 0.786 | 0.309 | 14.73 |

Table 3: **Ablation study with varying numbers of Gaussians.** We report PSNR (↑), SSIM (↑), LPIPS (↓), and memory usage in MB (↓), averaged across Mip-NeRF 360, Tanks&Temples, and DeepBlending. The top three results in each column are highlighted in red , orange , and yellow .

## 6 CONCLUSION

Textured Gaussians enhance visual quality by augmenting each Gaussian with an RGB texture map for spatially varying color and an alpha texture for spatially varying opacity, but these methods typically introduce too many parameters and causes memory inefficiency. To address this, we propose adaptive anisotropic textured Gaussians ($A^2TG$) that allows rectangular texture maps whose aspect ratios and resolutions are aligned with the geometry of the Gaussians. This reduces memory storage while preserving fine details. We also introduce a gradient-based adaptive texture control strategy to efficiently determine the texture shapes along with parameter update for joint optimization. $A^2TG$ has achieved best visual quality among state-of-the-art methods under the same memory constraint, demonstrating efficiency of our method.

**Limitation and future work.** We demonstrate that $A^2TG$ produces efficient textured Gaussians with better visual quality and memory efficiency. Currently, we only upscale texture resolution of Gaussians. To further improve the efficiency of texture usage, we can scale down the texture resolution when necessary. Moreover, compressing textures and controlling the shape and size of Gaussians can also improve memory efficiency. For future work, combining adaptive textures with flexible primitives beyond 2D/3D Gaussians such as Deformable Radial Kernel Splatting(Huang et al., 2025) might further enable better texture utilization since those primitives have flexible and sharper boundary shapes. In addition, we also plan to develop dynamic GS models, such as 4DGS, using $A^2TG$.

## ACKNOWLEDGEMENTS

The authors wish to thank our anonymous reviewers for their constructive feedback. The project was funded in part by the National Science and Technology Council of Taiwan (114-2221-E-007-114-MY3, 113-2221-E-007-102-MY3, 114-2221-E-007-115-MY3).

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

# A APPENDIX

## A.1 ADDITIONAL EXPERIMENTS COMPARING OUR METHOD WITH BASELINES.

Table 4 summarizes PSNR/SSIM/LPIPS and memory for all methods under fixed Gaussian budgets (#GS = 1M, 500k, 100k, 50k, and 10k) on Mip-NeRF 360, Tanks&Temples, and DeepBlending. Memory overheads are reported relative to the 2DGS baseline within each #GS block. Overall, texture-based methods consistently improve over non-textured 2DGS variants in visual quality. Among them, *Textured Gaussians\** and *BBSplat* often achieve the strongest metrics, but require substantially more memory: *Textured Gaussians\** is typically around +110% across budgets, while *BBSplat* becomes extremely heavy (about +1760%) at #GS=100k. In contrast, $A^2TG$ offers a better memory–quality trade-off, matching or approaching the best-performing textured baselines with far lower overhead. This advantage is most pronounced at higher Gaussian counts (e.g., 1M and 500k), where allocating high-resolution textures uniformly is most expensive and adaptive anisotropic textures provide the greatest benefit. At very low #GS (50k/10k), the relative memory gap narrows because the baseline footprint is already small; however, these regimes are less critical in practice since high-fidelity reconstructions typically rely on larger Gaussian budgets.

| Method | Mip-NeRF 360 | | | | Tanks & Temples | | | | DeepBlending | | | |
|---|---|---|---|---|---|---|---|---|---|---|---|---|
| | PSNR ↑ | SSIM ↑ | LPIPS ↓ | Mem (MB, +%) | PSNR ↑ | SSIM ↑ | LPIPS ↓ | Mem (MB, +%) | PSNR ↑ | SSIM ↑ | LPIPS ↓ | Mem (MB, +%) |
| *#GS = 1M* | | | | | | | | | | | | |
| 2DGS | – | – | – | – | – | – | – | – | 29.48 | 0.900 | 0.267 | 232 (0%) |
| 2DGS* | – | – | – | – | – | – | – | – | 29.58 | 0.899 | 0.263 | 232 (0%) |
| 2DGS*-MCMC | 28.37 | 0.838 | 0.172 | 232 (0%) | 23.11 | 0.827 | 0.148 | 232 (0%) | 29.60 | 0.898 | 0.190 | 232 (0%) |
| Super Gaussians | 28.39 | 0.848 | 0.218 | 296 (28%) | 23.66 | 0.847 | 0.185 | 284 (**22%**) | 29.38 | 0.905 | 0.253 | 296 (27%) |
| Textured Gaussians* | 28.81 | 0.846 | 0.159 | 488 (110%) | 23.58 | 0.833 | 0.141 | 488 (110%) | 29.64 | 0.897 | 0.185 | 488 (110%) |
| A²TG (Ours) | 28.70 | 0.844 | 0.163 | 291 (**25%**) | 23.58 | 0.831 | 0.145 | 292 (26%) | 29.82 | 0.900 | 0.180 | 277 (**19%**) |
| *#GS = 500k* | | | | | | | | | | | | |
| 2DGS | 27.38 | 0.805 | 0.283 | 116 (0%) | 22.91 | 0.819 | 0.234 | 116 (0%) | 29.24 | 0.894 | 0.286 | 116 (0%) |
| 2DGS* | 27.77 | 0.811 | 0.270 | 116 (0%) | 23.08 | 0.824 | 0.220 | 116 (0%) | 29.45 | 0.894 | 0.279 | 116 (0%) |
| 2DGS*-MCMC | 27.95 | 0.824 | 0.196 | 116 (0%) | 22.87 | 0.819 | 0.166 | 116 (0%) | 29.45 | 0.896 | 0.202 | 116 (0%) |
| Super Gaussians | 28.01 | 0.831 | 0.245 | 148 (28%) | 23.31 | 0.835 | 0.206 | 148 (28%) | 29.60 | 0.903 | 0.266 | 148 (28%) |
| Textured Gaussians* | 28.47 | 0.836 | 0.181 | 244 (110%) | 23.50 | 0.826 | 0.158 | 244 (110%) | 29.60 | 0.898 | 0.195 | 244 (110%) |
| A²TG (Ours) | 28.31 | 0.832 | 0.187 | 148 (**28%**) | 23.43 | 0.824 | 0.163 | 150 (29%) | 29.71 | 0.899 | 0.193 | 140 (**21%**) |
| *#GS = 100k* | | | | | | | | | | | | |
| 2DGS | – | – | – | – | 18.82 | 0.725 | 0.340 | 24 (0%) | 27.80 | 0.875 | 0.336 | 24 (0%) |
| 2DGS* | – | – | – | – | 19.17 | 0.734 | 0.324 | 24 (0%) | 28.32 | 0.879 | 0.324 | 24 (0%) |
| 2DGS*-MCMC | 26.40 | 0.767 | 0.294 | 24 (0%) | 22.07 | 0.776 | 0.245 | 24 (0%) | 28.69 | 0.883 | 0.253 | 24 (0%) |
| Super Gaussians | 23.12 | 0.737 | 0.366 | 30 (28%) | 20.11 | 0.737 | 0.327 | 30 (28%) | 26.61 | 0.875 | 0.331 | 30 (28%) |
| BBSplat | 27.73 | 0.828 | 0.216 | 433 (1764%) | 23.40 | 0.842 | 0.164 | 433 (1767%) | 28.78 | 0.894 | 0.268 | 433 (1764%) |
| Textured Gaussians* | 27.04 | 0.785 | 0.268 | 49 (110%) | 22.52 | 0.787 | 0.233 | 49 (111%) | 29.18 | 0.890 | 0.236 | 49 (110%) |
| A²TG (Ours) | 26.62 | 0.775 | 0.283 | 30 (**28%**) | 22.37 | 0.780 | 0.243 | 31 (32%) | 29.04 | 0.888 | 0.243 | 29 (**24%**) |
| *#GS = 50k* | | | | | | | | | | | | |
| 2DGS*-MCMC | 25.40 | 0.727 | 0.359 | 12 (0%) | 21.30 | 0.741 | 0.302 | 12 (0%) | 27.92 | 0.871 | 0.292 | 12 (0%) |
| BBSplat | 26.85 | 0.798 | 0.251 | 216 (1764%) | 22.68 | 0.803 | 0.225 | 216 (1764%) | 28.68 | 0.893 | 0.273 | 216 (1764%) |
| Textured Gaussians* | 26.08 | 0.748 | 0.329 | 25 (110%) | 21.78 | 0.756 | 0.285 | 25 (110%) | 28.59 | 0.880 | 0.271 | 25 (110%) |
| A²TG (Ours) | 25.64 | 0.736 | 0.347 | 15 (**26%**) | 21.62 | 0.747 | 0.297 | 16 (**31%**) | 28.26 | 0.876 | 0.283 | 15 (**24%**) |
| *#GS = 10k* | | | | | | | | | | | | |
| 2DGS*-MCMC | 22.81 | 0.618 | 0.536 | 3 (0%) | 19.39 | 0.632 | 0.473 | 3 (0%) | 25.13 | 0.822 | 0.436 | 3 (0%) |
| Textured Gaussians* | 23.41 | 0.638 | 0.501 | 5 (110%) | 20.05 | 0.654 | 0.442 | 5 (110%) | 25.97 | 0.835 | 0.400 | 5 (110%) |
| A²TG (Ours) | 23.09 | 0.628 | 0.518 | 3 (**19%**) | 19.79 | 0.645 | 0.456 | 3 (**25%**) | 25.50 | 0.827 | 0.421 | 3 (**18%**) |

Table 4: **Full multi-dataset evaluation.** Memory percentages are relative to **2DGS** at the same #GS. The top three results are highlighted in red, orange, and yellow, and the least parameter increases are in **bold**.

## A.2 ADDITIONAL DETAILS ON COMPUTATIONAL EFFICIENCY

3D Gaussian Splatting (Kerbl et al., 2023) is known for its real-time rendering capability. Our method builds upon 2D Gaussian Splatting (Huang et al., 2024) by augmenting each Gaussian with a non-uniform, variable-sized texture. During rendering, after computing the $uv$-coordinates of the intersection between a pixel and a Gaussian along the camera ray, we fetch texture values using bilinear interpolation.

To efficiently support non-uniform texture resolutions ranging from $\{1, 2, 4, \ldots, N^2\} \times \{1, 2, 4, \ldots, N^2\}$, where $N$ is the number of texture upscaling, we pack all per-Gaussian textures into a single large texture-atlas array located in GPU global memory. Each Gaussian stores additional metadata specifying its texture dimensions and its offset within the atlas. In our custom CUDA rasterization kernel, this structure results in up to 16 global-memory load operations per texture lookup, since bilinear interpolation may require up to four texel samples for each of the RGBA channels.

Although these texture fetches introduce additional overhead during both training and inference, the system still maintains real-time rendering speeds above 30 FPS. While we did not explicitly optimize the rasterizer for maximum throughput, further improvements are likely possible through vectorized texture loads or tile-level shared-memory prefetching.

Table 5 reports the inference FPS, per-frame inference time, and total training time for 2DGS (Huang et al., 2024), Textured Gaussians* (Chao et al., 2025), and A²TG on the DeepBlending dataset (Hedman et al., 2018a). Both A²TG and Textured Gaussians* incur additional computation compared to 2DGS due to texture sampling, which explains the decrease in inference FPS. Importantly, A²TG is consistently faster than Textured Gaussians* during both inference and training. The key reason is that A²TG allocates texture parameters adaptively rather than assigning a fixed-size texture to every Gaussian. This reduces the total number of texel reads and gradient updates, resulting in lower memory traffic and fewer texture operations. Consequently, A²TG achieves higher inference FPS and shorter training time than Textured Gaussians* under the same number of Gaussians.

| Method | 2DGS | A$^2$TG | Textured Gaussians* |
|---|---|---|---|
| Number of Gaussians | 500k | 500k | 500k |
| Model size (MB) | 116 | 140 | 244 |
| Inference FPS | 250 | 140 | 119 |
| Inference time per frame (s) | 0.004 | 0.007 | 0.008 |
| Training time (min) | 7.18 | 19.7 | 21.6 |

Table 5: **Computational Efficiency.** Comparison of the inference speed under the same number of Gaussians on the DeepBlending dataset. The render speed of A$^2$TG and Textured Gaussians is slower than 2DGS because of the extra texture sampling during rendering, while A$^2$TG has a slightly faster render speed than Textured Gaussians due to fewer texture parameters.

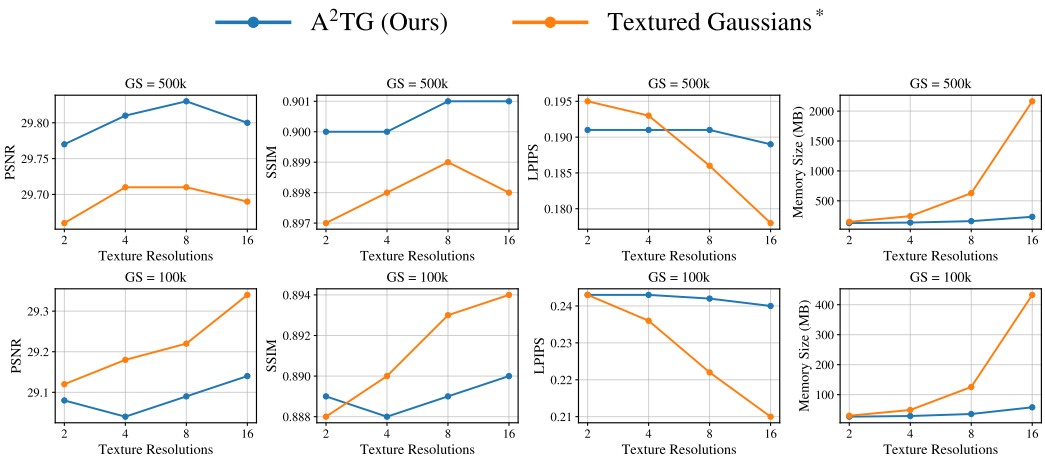

Figure 6: **Comparison of different texture resolutions with fixed Gaussian budgets on Deep-Blending.** Textured Gaussians* uses a fixed per-Gaussian texture size, while A$^2$TG adaptively allocates textures up to a maximum cap.

### A.3 ADDITIONAL EXPERIMENTS ON TEXTURE MAP RESOLUTION

To further study how texture resolution affects quality and memory, we compare Textured Gaussians* (Chao et al., 2025) with our A$^2$TG under multiple texture resolutions. For Textured Gaussians*, we sweep fixed per-Gaussian texture sizes from $2\times2$ to $16\times16$. For A$^2$TG, we evaluate the corresponding maximum texture caps (Max $2\times2$ to Max $16\times16$), where textures are adaptively allocated up to the specified cap.

Table 6 and Figure 6 summarize results under fixed Gaussian budgets (100k and 500k Gaussians). Overall, A$^2$TG provides a strong memory–quality trade-off compared to Textured Gaussians*: it matches or improves PSNR/SSIM and reduces LPIPS while using less memory at lower texture resolutions. At higher texture resolutions, Textured Gaussians* can achieve slightly better quality, but the memory cost grows rapidly and becomes prohibitively expensive. Moreover, increasing texture resolution yields diminishing returns in visual metrics, which we attribute to our gradient-based texture upscaling procedure repeatedly prioritizing the same subset of Gaussians. In practice, these trends suggest that a maximum texture cap of $2\times2$ or $4\times4$ is most appropriate for A$^2$TG. Overall, the results highlight the benefit of adaptive texture allocation: it concentrates resolution where it is most beneficial while avoiding the uniform memory growth incurred by fixed-resolution textures.

### A.4 DECLARATION OF LLM USAGE

We used large language models (LLMs) solely for polishing the text in this paper. Specifically, LLMs assisted in refining grammar, improving readability, and adjusting tone for academic writing.

| Method | Tex. | PSNR ↑ | SSIM ↑ | LPIPS ↓ | Mem (MB) ↓ |
|---|---|---|---|---|---|
| *#GS = 500k* | | | | | |
| Textured Gaussians* | 2×2 | 29.66 | 0.897 | 0.195 | 148.00 |
| Textured Gaussians* | 4×4 | 29.71 | 0.898 | 0.193 | 244.00 |
| Textured Gaussians* | 8×8 | 29.71 | 0.899 | 0.186 | 628.00 |
| Textured Gaussians* | 16×16 | 29.69 | 0.898 | 0.178 | 2164.00 |
| A$^2$TG | Max 2×2 | 29.77 | 0.900 | 0.191 | 127.66 |
| A$^2$TG | Max 4×4 | 29.81 | 0.900 | 0.191 | 136.24 |
| A$^2$TG | Max 8×8 | 29.83 | 0.901 | 0.191 | 159.95 |
| A$^2$TG | Max 16×16 | 29.80 | 0.901 | 0.189 | 232.82 |
| *#GS = 100k* | | | | | |
| Textured Gaussians* | 2×2 | 29.12 | 0.888 | 0.243 | 29.60 |
| Textured Gaussians* | 4×4 | 29.18 | 0.890 | 0.236 | 48.80 |
| Textured Gaussians* | 8×8 | 29.22 | 0.893 | 0.222 | 125.60 |
| Textured Gaussians* | 16×16 | 29.34 | 0.894 | 0.210 | 432.80 |
| A$^2$TG | Max 2×2 | 29.08 | 0.889 | 0.243 | 26.44 |
| A$^2$TG | Max 4×4 | 29.04 | 0.888 | 0.243 | 28.68 |
| A$^2$TG | Max 8×8 | 29.09 | 0.889 | 0.242 | 35.39 |
| A$^2$TG | Max 16×16 | 29.14 | 0.890 | 0.240 | 57.77 |

Table 6: **Comparison of different texture resolutions with fixed Gaussian budgets on Deep-Blending.** Textured Gaussians* uses a fixed per-Gaussian texture size, while A$^2$TG adaptively allocates textures up to a maximum cap.

All research ideas, methodology, analyses, results, and terminology definitions presented in this work are original contributions from the authors and were not generated by LLMs.

