# OpenReview forum: "A^2TG: Adaptive Anisotropic Textured Gaussians for Efficient 3D Scene Representation"
_ICLR.cc/2026/Conference — ICLR 2026 Poster_

### Official Review · Reviewer_Q3gj · 2025-10-28

**Soundness:** 3
**Presentation:** 3
**Contribution:** 2
**Rating:** 4
**Confidence:** 4

**Summary:**

The paper targets the inefficiency of assigning uniformly sized, square per-primitive textures in Gaussian-splatting pipelines. The authors propose A2TG on top of 2DGS: (1) start from a trained 2DGS model and attach per-Gaussian RGBA texture patches; (2) tie texture upscaling and aspect ratio to positional-gradient signals; (3) implement a lightweight, discrete allocation over {1,2,4}×{1,2,4} with a thresholded trigger and a periodic upscaling schedule. Experiments under fixed-memory and fixed-#Gaussians protocols on Mip-NeRF360, Tanks&Temples, and DeepBlending show comparable or better rendering quality at lower memory.

**Strengths:**

- This paper is well-written and easy to understand.
- This paper introduces anisotropic textures and achieves a slight reduction in memory footprint without compromising rendering quality.
- The authors tie texture resolution to the gradient, enabling detail-aware texture optimization, which makes sense.
- The ablations are appreciated to show the contribution of each part.

**Weaknesses:**

- Missing references for some important related works:
  - [MM 2024] AbsGS: Recovering Fine Details for 3D Gaussian Splatting, by Zongxin Ye et al. AbsGS, similar to the `absolute value of its positional gradient` described in L229–230, is the key mechanism by which GS-based methods eliminate floating floaters.
  - HDGS: Textured 2D Gaussian Splatting for Enhanced Scene Rendering, by Yunzhou Song et al. Like the present work, HDGS adopts a Texture-GS framework built on 2DGS, and therefore needs detailed discussion and comparison in the **Related Work** section.
- While I appreciate the core idea proposed by the authors, the ablation in Table 3 suggests that the upscaling strategy and anisotropic textures contribute only marginally to the rendering quality. Their benefits are reflected primarily in reduced memory overhead. The improvements reported in Table 1 likely stem more from the inherently efficient memory management of 2DGS and the densification mechanism of 3DGS-MCMC.
- There is no visualization of the textures. I understand it may be difficult to visualize textures at different levels of detail, but I do not think rendering quality is the primary aspect to focus on when introducing texture maps. We should care more about the quality of the textures themselves.
- Also, no comparison on FPS and training time.
- Minor typo errors:
  - L53: `As the results` -> `As a result`
  - L81: `build upon 3DGS` -> `builds upon 3DGS`
  - L184-185: `are now calculated` -> `is now calculated`
  - L197: `the pixel and alpha value` -> `the pixel and alpha values`
  - L261: `the memory size of each algorithms resulting parameters`

**Questions:**

1. Could you specify the exact values of the thresholds (e.g., the anisotropy thresholds **kA, kS**)?
2. What motivates restricting texture sizes to the discrete set **{1,2,4}×{1,2,4}**? Did you try larger grids or continuous allocation, and how do they affect quality vs. memory?

**Details Of Ethics Concerns:**

None.

---

> ### Author Response · Authors · 2025-11-28
> **Reply to Reviewer Q3gj**
>
> ### **1. Missing references. (Weakness 1)**
> Thank you for pointing this out. We have added the missing references for AbsGS at lines 237-238 and included HDGS in the related works section at lines 96-98.
>
> ### **2. Concerns regarding the contribution of our upscaling method compared with the MCMC. (Weakness 2)**
> Both our A$^2$TG method and Textured Gaussians extend 2DGS with the MCMC optimization strategy. Under the same memory constraints, however, A$^2$TG achieves higher rendering quality than Textured Gaussians. Although the improvements in Table 3 appear moderate, increasing the number of 2D Gaussian points to match the memory overhead of additional textures would yield only marginal visual gains—typically smaller than the gains achieved through adaptive texture parameters. Too better demonstrate the results, we’ve added a new baseline 2DGS*-MCMC to the revised paper (table 1, table 2).
>
> Below is the Mip-NeRF 360 part of table 1 in the revised paper:
> | Method              | PSNR ↑ | SSIM ↑ | LPIPS ↓ | #GS  | Mem  |
> |---------------------|--------|--------|---------|------|-------|
> | 2DGS*               | 26.72  | 0.787  | 0.308   | 259k | 60MB  |
> | 2DGS*-MCMC          | 27.47  | 0.806  | 0.228   | 259k | 60MB  |
> | Super Gaussians     | 25.29  | 0.791  | 0.300   | 202k | 60MB  |
> | BBSplat             | 23.07  | 0.614  | 0.462   | 14k  | 60MB  |
> | Textured Gaussians* | 27.02  | 0.789  | 0.261   | 123k | 60MB  |
> | A²TG (Ours)         | 27.64  | 0.809  | 0.230   | 207k | 57MB  |
>
> This observation is also reflected in Table 1: under identical memory budgets (e.g., 60 MB), A$^2$TG consistently delivers the best performance across all methods. This highlights that adaptively allocating texture resolution is more effective for improving novel view synthesis quality than simply increasing the number of 2D Gaussians (training 2DGS with more point clouds) or assigning a fixed-size texture to every Gaussian, as done in Textured Gaussians*.
>
> ### **3. Visualization on textures and texture quality (Weakness 3)**
> We thank the reviewer for raising this point. In the revised paper, we have added a dedicated analysis (Section 5.2 Comparisons: Texture Decomposition) to examine *what* the adaptive textures learn. Rather than visualizing the raw texture maps—which are difficult to interpret because textures differ in resolution, spatial locality, and illumination context—we instead evaluate the quality of the learned textures through a targeted ablation study.
>
> As suggested, we focus on assessing the *quality of the textures themselves*, not merely the final rendering. In the revised supplementary material (Section 5.2, Fig. 5), we render each scene under three conditions:
>
> 1. **Full A$^2$TG rendering** (SH base color + adaptive textures),
> 2. **Without textures** (RGB textures set to zero; alpha textures set to one), and
> 3. **Without base color** (SH color removed; appearance determined solely by the textures).
>
> This visualization clearly shows that:
> - the **textures learn high-frequency, view-consistent residual details** such as foliage structure and fabric patterns,
> - the **SH base color provides low-frequency shading and global structure**, and
> - their **combination** produces the final photorealistic result.
>
> We believe this form of analysis is more meaningful and interpretable than displaying standalone texture grids, whose appearance is tied to each Gaussian’s local coordinate frame and coverage. We appreciate your feedback, and the revised version now includes detailed experiments that directly evaluate the quality and role of the learned textures.
>
>
> ### **4. Concerns about render speed and training time experiments of non-uniform Textured Gaussians. (Weakness 4)**
> Thank you. We conducted additional experiments to measure rendering speed and found that A$^2$TG achieves real-time performance (140 FPS with 500k Gaussians) and competitive training times, despite incorporating anisotropic textures. Notably, our method renders and trains faster than Textured Gaussians despite using anisotropic textured Gaussians, as our design effectively controls the necessary texture resolution of these primitives. We have included the updated results in the revision and provided further analysis in the revised supplementary material (Section A.2, Table 6).
>
> Table 6:
> | Model | 2DGS | A2TG (Ours) | Textured Gaussians* |
> | :--- | :---: | :---: | :---: |
> | Number of Gaussians | 0.50 M | 0.50 M | 0.50 M |
> | Model size (MB) | 116 MB | 140 MB | 244 MB |
> | Inference FPS | 250 | 140 | 119 |
> | Inference time per frame (s) | 0.004 | 0.007 | 0.008 |
> | Training time (min) | 7.18 | 19.7 | 21.6 |

---

> ### Author Response · Authors · 2025-11-28
> **Reply to Reviewer Q3gj**
>
> ### **5. Clarification on thresholds .**
> Thank you. All results reported in our paper use the same fixed parameters across all datasets: $k_A = 4.0$, $k_S = 0.01$, and $k_G = 0.00002$. We have clarified in our revised paper (Lines 317-319).
>
> ### **6. Questions on the choice of discrete texture resolutions and experiments with continuous texture sizes.**
> Thank you. Our choice to use only texture resolutions that are powers of two is primarily empirical. Since we apply nearest-neighbor sampling for texture upscaling, power-of-two resolutions guarantee that the upscaled texture preserves the exact appearance of the original texture while increasing its grid density. This avoids interpolation artifacts and maintains consistency across scales.
>
> We conducted preliminary experiments on *continuous* texture allocation. In these experiments, **A$^2$TG** contains texture resolutions from the discrete set {1,2,4}x{1,2,4}, whereas **A$^2$TG-Increment** uses a finer resolution space {1,2,3,4}x{1,2,3,4}. Specifically, A$^2$TG-Increment performs texture upscaling by gradually incrementing the resolution over three rounds of texture upscaling. Experiments on the MipNeRF-360 dataset show that both variants produce nearly identical reconstruction quality.
>
> | Method             | PSNR  | SSIM  | LPIPS | Mem (MB) | #GS |
> |--------------------|-------|-------|-------|----------|-----|
> | A$^2$TG-Increment     | 23.01 | 0.625 | 0.525  | 2.65     | 10k |
> | A$^2$TG               | 23.02 | 0.625 | 0.524 | 2.66     | 10k |
>
>
> We also explored the effect of allowing **larger maximum texture grids**. The following experiments, performed on the MipNeRF-360 dataset with maximum texture sizes ranging from $2×2$ to $16×16$, indicate that larger textures correlate with slightly higher PSNR. However, the improvement remains small relative to the additional parameters. We further observe that the gradient-based upscaling policy consistently selects the *same* subset of Gaussians to upscale—typically those corresponding to scene details well-covered by many training views. Such regions generally do not benefit from excessively large texture allocations since they contain relatively small Gaussian, explaining the diminishing returns from larger grids.
>
> | Method                 | PSNR  | SSIM  | LPIPS | Mem (MB) | #GS |
> |------------------------|-------|-------|-------|----------|-----|
> | A$^2$TG (Max 2×2)         | 25.61 | 0.735 | 0.348 | 12.83    | 50k |
> | A$^2$TG (Max 4×4)         | 25.62 | 0.735 | 0.347 | 14.25    | 50k |
> | A$^2$TG (Max 8×8)         | 25.65 | 0.739 | 0.340 | 19.40    | 50k |
> | A$^2$TG (Max 16×16)       | 25.70 | 0.741 | 0.331 | 38.43    | 50k |
>
> ### **7. Typo corrections.**
> We appreciate the reviewer’s attention to detail and have corrected the identified typos.

---

### Official Review · Reviewer_Cw5H · 2025-10-29

**Soundness:** 3
**Presentation:** 3
**Contribution:** 3
**Rating:** 6
**Confidence:** 1

**Summary:**

The paper proposes Adaptive Anisotropic Textured Gaussians (A2TG), which equips each Gaussian primitive with an anisotropic, gradient-adaptive texture, improving memory efficiency and visual fidelity over fixed-square textures in real-time 3D scene rendering.

**Strengths:**

- The paper introduces Adaptive Anisotropic Textured Gaussians (A2TG), a novel extension of textured Gaussian splatting that assigns each Gaussian an anisotropic texture whose resolution and aspect ratio are adaptively determined.
- The paper is well-structured, with a clear motivation for adaptive texture allocation, detailed explanations of the method, and thorough evaluation of results.

**Weaknesses:**

While the proposed A2TG effectively improves texture allocation and memory efficiency, the method relies on several heuristically chosen thresholds for gradient-driven selection and anisotropy-based upscaling, which may require dataset-specific tuning. Additionally, the iterative upscaling procedure introduces extra computational overhead, potentially limiting real-time performance for very large or highly detailed scenes. Finally, the approach does not explicitly handle occlusion interactions beyond local gradients, which could affect texture fidelity in heavily occluded regions.

The paper has some typos:
- Line 261, "LPIPs"
- Line 314, "... across varies textured-based 2DGS...."
- Line 322, "... latter having more nubmer"
- Line 364, "the visaul quality"
- Line 368-369, "Textured Guassians*"
- Line 428, " (w/o Uscaling)"
- Line 435-436, "methods typical introduce"
- Line 441, "among the-state-of-the-art methods"

**Questions:**

See Weaknesses

---

> ### Author Response · Authors · 2025-11-28
> **Reply to Reviewer Cw5H**
>
> ### **1. Concerns about dataset-specific tuning of the parameters. (Weakness 1)**
> Thank you for the question. All results in our paper use the same fixed parameters across all datasets—$k_A = 4.0$, $k_S = 0.01$, and $k_G = 0.00002$ (Lines 318–319)—without any dataset-specific tuning. Even under this fixed setting, our method consistently achieves visual quality comparable to Textured Gaussians* while using significantly fewer parameters (table 1, table 2).
>
>
>
> ### **2. Concerns about the render speed of anisotropic textured Gaussians. (Weakness 2)**
> Thank you. We conducted additional experiments to measure rendering speed and found that A$^2$TG achieves real-time performance (140 FPS with 500k Gaussians). Notably, our method renders faster than Textured Gaussians despite using anisotropic textured Gaussians, as our design effectively controls the necessary texture resolution of these primitives. We have included the updated results in the revision and provided further analysis in the revised supplementary material (Section A.2, Table 6).
>
> Table 6:
> | Model | 2DGS | A$^2$TG (Ours) | Textured Gaussians* |
> | :--- | :---: | :---: | :---: |
> | Number of Gaussians | 0.50 M | 0.50 M | 0.50 M |
> | Model size (MB) | 116 MB | 140 MB | 244 MB |
> | Inference FPS | 250 | 140 | 119 |
> | Inference time per frame (s) | 0.004 | 0.007 | 0.008 |
> | Training time (min) | 7.18 | 19.7 | 21.6 |
>
> ### **3. Concerns about handling occlusion interactions.**
> Thank you for pointing this out. Our gradient-based texture upscaling strategy implicitly assigns fewer texture parameters to Gaussians that are frequently occluded, because they accumulate smaller gradients. While it is possible that some heavily occluded Gaussians may not receive enough texture resolution to fully match fine details, allocating more texture parameters to them risks overfitting, since they are supervised by only a limited number of visible views. Determining the optimal balance in such cases is non-trivial and difficult to evaluate conclusively.
>
> ### **4. Typo corrections.**
> We appreciate the reviewer’s attention to detail and have corrected the identified typos.

---

### Official Review · Reviewer_QVPU · 2025-10-29

**Soundness:** 3
**Presentation:** 3
**Contribution:** 3
**Rating:** 6
**Confidence:** 4

**Summary:**

This paper proposes Adaptive Anisotropic Textured Gaussians (A2TG) to enhance texture representations by extending fixed square textures to adaptively allocated anisotropic resolutions, reducing memory consumption while improving image quality.
The resolution and aspect ratio are controlled by an adaptive gradient-based strategy during optimization.

**Strengths:**

1. This paper clearly identifies the problem of memory redundancy in textured Gaussians.
2. The proposed anisotropic textures demonstrate good results in terms of storage efficiency.
3. Using gradients to guide texture scaling is clear and well-motivated.
4. The paper is well-organized and readable.

**Weaknesses:**

1. Some experiments could be added for further validation:
1.1. The gradient-based densification follows the spirit of the adaptive density control procedure in 3DGS. Experiments re-enabling similar densification procedures in 2DGS or baseline models are desired and could further demonstrate the effectiveness of the proposed method.
1.2. Since the performance of Textured Gaussians also depends on the resolution of textures, how does the proposed method perform compared with Textured Gaussians using 2×2 resolutions? Would the proposed method still achieve consistent advantages?
1.3. What about results with more Gaussians (e.g. 1M)?
2. Some mathematical notations are unclear, making it difficult to evaluate the correctness of the equations.
2.1. For example, the c_{\cdot}(x_j) notations in Eq.6 and Eq.7 refer to different concepts but lack explanations. Could the authors update the equations consistently for easier evaluation? From the current version, I suspect Eq. 7 may correspond to dL/d\alpha instead of dc/d\alpha. Since the labels are used interchangeably, a consistent version with clear explanations would help the final evaluation.
The second bullet on line 242: should it be r < 1/k_A?
2.2. Various typos and hard-to-understand sentences remain; see Questions below.

**Questions:**

1. There are some contradictory claims about image-quality improvements (lines 53, 55–56, 66–67, and the Experiments section). Considering the proposed method is a special case of the fixed square resolution (albeit with more parameters), why does it improve image quality?
2. How is the number of 100 K Gaussians guaranteed, since SfM initialization may exceed 100 K? Do you use low-resolution images or other strategies?
3. According to the logic from lines 239–244, does the upscaling double the resolution when r is within [1/K_A, K_A], regardless of s_x or s_y values? This seems to contradict intuition.
4. Does the anisotropic texture introduce runtime overhead?

Typos
1. Use "texture-based" instead of "textured-based".
2. Line 314: replace “varies” with “various.”
3. Line 318: change “resulting” to “resulting in,” and add “a” before “fixed.”
4. Lines 322–323: the sentences are fragmented and hard to follow—please clarify why A2TG is the best.
5. Line 367, 368, then to than

---

> ### Author Response · Authors · 2025-11-28
> **Reply to Reviewer QVPU**
>
> ### **1. Clarification on re-enabling similar densification procedures in 2DGS or baseline models. (Weakness 1.1)**
> Thank you. Jointly optimizing both the number of Gaussians and the texture resolution, similar to the densification procedure in 2DGS, significantly increases optimization complexity because the system must decide when to densify and when to upscale textures based on gradient thresholds. Our preliminary experiments indicate that this joint optimization is much harder to stabilize in practice. For this reason, we adopt a two-stage training strategy that separates density control from texture upscaling, which results in more stable training and better overall efficiency.
>
>
> ### **2. Additional experiments on 2x2 texture sizes. (Weakness 1.2)**
> Thank you for the suggestion. We conducted additional experiments comparing our method with Textured Gaussians*. For both our method and Textured Gaussians*, we evaluate texture resolutions of 2×2 and 4×4. We report results on the Mip-NeRF 360 dataset with the number of Gaussians fixed at 50k (see Table 7 below). As shown, A$^2$TG with a maximum texture resolution of 2×2 achieves comparable PSNR/SSIM/LPIPS to Textured Gaussians* with 2×2, while using less memory. We have included these results in the revised supplementary material (Section A.3, Table 7).
>
> Table 7:
> | Method           | PSNR  | SSIM  | LPIPS | Mem (MB) | #GS |
> |------------------|-------|-------|-------|----------|-----|
> | Textured Gaussians* (2×2)         | 25.59 | 0.734 | 0.347 | 14.80    | 50k |
> | Textured Gaussians* (4×4)         | 25.77 | 0.741 | 0.338 | 24.40    | 50k |
> | A$^2$TG (Max 2×2)   | 25.62 | 0.735 | 0.348 | 12.83    | 50k |
> | A$^2$TG (Max 4×4)   | 25.63 | 0.733 | 0.350 | 14.09    | 50k |
>
> ### **3. Results with more Gaussians (e.g., 1M). (Weakness 1.3)**
> Thank you. We have added the results for 1M Gaussians on the DeepBlending dataset. These experiments show that our method uses significantly less memory than Textured Gaussians*. The updated results are included in the revised supplementary file (Section A.1, Table 5).
>
> Table 5:
> | Method              | PSNR  | SSIM  | LPIPS | Mem (MB) | #GS |
> |---------------------|-------|-------|-------|----------|-----|
> | 2DGS*               | 29.58 | 0.898 | 0.190 | 232      | 1M  |
> | Super Gaussians     | 29.38 | 0.905 | 0.253 | 295      | 1M  |
> | Textured Gaussians* | 29.66 | 0.896 | 0.183 | 488      | 1M  |
> | A$^2$TG                | 29.55 | 0.894 | 0.190 | 266      | 1M  |

---

> ### Author Response · Authors · 2025-11-28
> **Reply to Reviewer QVPU**
>
> ### **4. Clarification on the claim of image quality improvements. (Question 1)**
> Thank you. A$^2$TG provides efficient memory usage while maintaining visual quality comparable to Texture-Gaussians*. We have clarified this consistently in the revised paper (Lines 53, 55-57, 65-67).
>
> ### **5. Explanation on achieving exactly 100k Gaussians. (Question 2)**
> For the 2DGS results in Table 2 under the 100k setting, the initial SfM point clouds of the Tanks & Temples and DeepBlending datasets contain fewer than 100k points. For the Mip-NeRF360 dataset, several scenes have more than 100k initial points, which is why those entries in the #GS = 100k section remains empty.
>
> For the other baseline methods, their implementations remove some of the initial SfM point clouds when they exceed the target number of Gaussians. As a result, these methods can always produce exactly 100k Gaussians for all scenes.
>
>
> ### **6. Clarification on texture upscaling. (Weakness 2.1 & Question 3)**
> Thank you. Our goal is to apply texture upscaling to Gaussians whose gradient magnitude exceeds a specified threshold, independent of their overall size. However, when a Gaussian is highly elongated, full 2D upscaling becomes unnecessary. To handle such cases, we first examine the anisotropy ratio $s_x / s_y$ and $s_y / s_x$, which indicates whether the Gaussian exhibits extreme elongation.
>
> This ratio alone is insufficient because a Gaussian can have a large anisotropy value while still having a relatively wide minor axis (i.e., spanning multiple pixels), in which case restricting upscaling to only one direction is not desirable. Therefore, our method applies **1D upscaling only when two conditions are met simultaneously**:
> (1) the Gaussian is highly anisotropic, and
> (2) its thinner axis is sufficiently small (i.e., covers fewer than a few pixels).
>
> Only Gaussians that satisfy both criteria are restricted to 1D upscaling; otherwise, we apply full 2D upscaling. We have revised the notation in the revised paper (Lines 243-250).
>
> ### **7. Concern about render speed of anisotropic Textured Gaussians. (Question 4)**
> Thank you. We conducted additional experiments to measure rendering speed and found that A$^2$TG achieves real-time performance (140 FPS with 500k Gaussians). Notably, our method renders faster than Textured Gaussians despite using anisotropic textured Gaussians, as our design effectively controls the necessary texture resolution of these primitives. We have included the updated results in the revision and provided further analysis in the revised supplementary material (Section A.2, Table 6).
>
> **Table 6:**
> | Model | 2DGS | A2TG (Ours) | Textured Gaussians* |
> | :--- | :---: | :---: | :---: |
> | Number of Gaussians | 0.50 M | 0.50 M | 0.50 M |
> | Model size (MB) | 116 MB | 140 MB | 244 MB |
> | Inference FPS | 250 | 140 | 119 |
> | Inference time per frame (s) | 0.004 | 0.007 | 0.008 |
> | Training time (min) | 7.18 | 19.7 | 21.6 |

---

> ### Author Response · Authors · 2025-11-28
> **Reply to Reviewer QVPU**
>
> ### **8. Clarification on gradient calculation notations (Weakness 2.1).**
> Thank you for pointing out the ambiguous notations. The main source of confusion was that we used bold $\mathbf{c}$ to denote the alpha-blended pixel color and $\mathbf{c}$_k to denote the $k$th pixel color, while also using $c_i$ to denote the per-Gaussian color. We have revised the notation in the gradient derivations (Equation 6) to clearly separate these two quantities and stay consistent with Equation 5.
>
> In the updated notation:
> - $\mathbf{c}$ denotes the RGB pixel color,
> - $\mathbf{c}^k$ denotes the $k$-th color channel of the rendered pixel, and
> - $c_i$ (non-bold) denotes the color parameter of the $i$-th 2D Gaussian.
>
> This separation ensures that pixel-level and Gaussian-level colors cannot be confused in the gradient expressions.
>
> Therefore in equation(6) of the paper:
> $$\frac{\partial \mathcal L_j}{\partial \mu_{i,x}}  = \sum_{k=1}^{3} \frac{\partial \mathcal L_j}{\partial {\bf c}^k({\bf x}_j)} \cdot \frac{\partial {\bf c}^k({\bf x}_j)}{\partial \alpha_i({\bf x}_j)} \cdot \frac{\partial \alpha _i({\bf x}_j)}{\partial \mu _{i,x}}.$$
>
> This is the gradient of the loss function of the $j$th pixel with respect to the $x$ coordinate of the position of the $i$th Gaussian. The three terms on the right-hand side represent these quantities separately:
>
> 1. Partial derivative of the loss function of the $j$th pixel with respect to the alpha-blended color at $j$th pixel.
> 2. Partial derivative of the alpha-blended color at $j$th pixel with respect to the alpha value of the $i$th Gaussian at camera ray $\mathbf{x}_j$($j$th pixel).
> 3. Partial derivative of the alpha value of the $i$th Gaussian at camera ray $\mathbf{x}_j$($j$th pixel) with respect to the $x$ coordinate of the position of the $i$th Gaussian.
>
> ### **9. Typo corrections.**
> We appreciate the reviewer’s attention to detail and have corrected the identified typos.

---

### Official Review · Reviewer_GiPV · 2025-11-01

**Soundness:** 3
**Presentation:** 3
**Contribution:** 3
**Rating:** 6
**Confidence:** 5

**Summary:**

The paper introduces an enhanced 2D Gaussian splatting technique that incorporates texture maps with adaptively controlled resolutions. The intersection point of a ray with the tangent plane of the 2D Gaussian splat is computed within the CUDA rendering pipeline. Using the tangent plane coordinates (uv), the corresponding texture color is retrieved from the texture map. The resolution of the texture maps is dynamically adjusted based on the accumulated gradients of the local uv coordinates. This approach supports 'anisotropic' resolutions, enabling different levels of detail along the u and v axes. Experimental results demonstrate that the proposed method achieves superior rendering quality compared to state-of-the-art Textured Gaussians, while also reducing storage requirements.

**Strengths:**

- Vanilla 3DGS and 2DGS require a large number of primitives to capture the high-frequency appearance and geometry of 3D scenes. This limitation arises from their restricted texture representation, which uses only a single color from a fixed viewpoint. Textured Gaussians, such as GStex, HDGS, Billboard Splatting, and SuperGaussians, represent significant advancements in this domain, enhancing Gaussian splatting with textures for improved 3D reconstruction. However, these methods do not account for the adaptive control of GS textures, leading to inefficiencies, as different primitives require varying levels of texture detail. Building on this insight, this paper introduces adaptive textures for GS, enabling variable resolutions for each primitive and along different axes. This approach is both innovative and practical.

- The results are impressive and highly satisfactory. As demonstrated in Tables 1 and 2 and Figure 2, A^2GS achieves superior performance with reduced storage requirements thanks to its adaptive control of GS texture resolutions. Additionally, Figure 3 highlights the method’s superiority in reconstructing fine details.

**Weaknesses:**

The first weakness lies in the methodology and is quite fundamental. The paper does not clarify how textures are fetched based on UV coordinates. Is it through bilinear interpolation or nearest neighbor? If bilinear interpolation is used, Equation 6 omits the terms \(\frac{\partial c_l(x_j)}{\partial \mu_{i,x}}\) and \(\frac{\partial c_l(x_j)}{\partial \mu_{i,y}}\), as \(c_l(x_j)\) is linearly related to the UV coordinates. On the other hand, if bilinear interpolation is not used, the optimization process and model capability are constrained. Since the positions of Gaussian splats (GS) are jointly optimized, if the gradients of textures are not related to the intersection coordinates, the texture map gradients cannot effectively guide GS movements for better fitting. Furthermore, using nearest neighbor (NN) queries results in discontinuous textures, which could reduce overall quality.

The second weakness concerns the experiments. The paper does not evaluate rendering speed, an essential aspect of the approach. Because GS textures have varying resolutions, the indexing of each GS texture is non-uniform. As a result, within each CUDA grid, it is impossible to define a fixed-length shared array for threads within the grid and move the data in parallel from the global to the shared. Consequently, textures must be fetched from the global CUDA memory, which is slower. Additionally, calculating the cumulative sum of each GS texture resolution to obtain the global base index for each GS texture adds computational overhead. These factors likely impact rendering speed and should be evaluated to demonstrate the tradeoff between performance and speed. While a reduction in speed is acceptable to an extent, maintaining real-time performance (>30 FPS) is crucial. If the speed drops below this threshold, the limitation should be explicitly acknowledged as a significant constraint.

The third weakness pertains to the conclusion section. A visible but minor limitation is that the method does not address reducing texture resolution dynamically. For instance, if a GS is optimized to have finer textures during an incorrect stage of the process but later becomes smaller or moves to textureless regions, the high-resolution texture may no longer be necessary. Addressing this issue could further improve efficiency. Additionally, while the proposed method makes substantial contributions to GS textures, it raises the possibility of combining adaptive textures with flexible 2D primitives, such as Deformable Radial Kernel Splatting. This combination could enable both flexible boundary shapes and sharpness while preserving richer texture representations. These two points should be considered as limitations and opportunities for future work.

**Questions:**

My suggestions are as follows: (1) provide a detailed explanation of the texture color query process using UV coordinates; (2) include an evaluation of the rendering speed; and (3) revise the conclusion section. I believe addressing these points will significantly enhance the quality of the paper, and I would be inclined to raise the score. While I acknowledge the novelty and contribution of the paper, improving these aspects is essential to further strengthen its overall quality.

---

> ### Author Response · Authors · 2025-11-28
> **Reply to Reviewer GiPV**
>
> ### **1. Clarification on Texture Fetching via UV Coordinates (Weakness 1)**
>
> Thank you for your insightful comments and observations. We use bilinear interpolation for querying texture values from UV coordinates. This ensures smooth gradients and correct differentiability. We have explicitly clarified the use of bilinear interpolation in the revised manuscript (Lines 192–195).
>
> ### **2. Analysis of Rendering Speed for Anisotropic Textured Gaussians (Weakness 2)**
>
> Thank you. We conducted additional experiments to measure rendering speed and found that $A^2TG$ achieves real-time performance (140 FPS with 500k Gaussians). Notably, our method renders faster than Textured Gaussians* despite using anisotropic textured Gaussians, as our design effectively controls the necessary texture resolution of these primitives. We have included the updated results in the revision and provided further analysis in the revised supplementary material (Section A.2, Table 6).
>
> **Table 6**: Computational Efficiency: Comparison of the inference speed under the same number
> of Gaussians on the DeepBlending dataset. The render speed of A$^2$TG and Textured Gaussians is
> slower than 2DGS because of the extra texture sampling during rendering, while A$^2$TG has a slightly
> faster render speed than Textured Gaussians due to fewer texture parameters.
>
> | Model | 2DGS | A$^2$TG (Ours) | Textured Gaussians* |
> | :--- | :---: | :---: | :---: |
> | Number of Gaussians | 0.50 M | 0.50 M | 0.50 M |
> | Model size (MB) | 116 MB | 140 MB | 244 MB |
> | Inference FPS | 250 | 140 | 119 |
> | Inference time per frame (s) | 0.004 | 0.007 | 0.008 |
> | Training time (min) | 7.18 | 19.7 | 21.6 |
>
> **Implementation Details & Performance Trade-off:**
>
> To handle anisotropic textures, we consolidate all individual texture maps into a single global texture atlas array. During fetching, we use precalculated per-Gaussian dimensions and offsets to sample from this atlas via bilinear interpolation. While calculating these offsets incurs a slight overhead in instruction count compared to uniform textures, $A^2TG$ requires significantly fewer total texture parameters. Although runtime performance is not the primary focus of this work, we believe that further engineering improvements, such as shared memory prefetching and vectorized texture memory access, could yield even greater performance gains. We have discussed this experiment in more detail in the revised supplementary material (Section A.2).
>
> ### **3. Future Work: Texture Downscaling and Flexible Primitives (Weakness 3)**
>
> We appreciate these valuable suggestions. In the revised Conclusion (Lines 502-508), we have expanded our discussion on future research directions. Specifically, we address the potential for dynamically downscaling texture resolution during optimization to further enhance efficiency. Additionally, we discuss the promising direction of combining adaptive textures with more flexible 2D primitives (such as Deformable Radial Kernel Splatting) to achieve greater flexibility in texture boundary shapes and sharpness.

---

### Author Response · Authors · 2025-11-28
**General Response**

We thank all reviewers for their insightful comments and constructive feedback. We are glad that you consider A$^2$TG innovative and practical for the problem of memory redundancy in textured Gaussians. Reviewers also highlighted that our gradient-guided texture scaling is well-motivated, the results achieve strong performance with reduced storage.

In this revision, we have addressed the concerns regarding our expositions, experiments, rendering time, and implementation details.

More specifically, we have made the following revisions:
1. Updated the claims in the **Introduction** for clarity and consistency.
2. Added missing references in the **Related Work** and **Methodology** sections.
3. Refined and clarified the notations used in the gradient derivation and texture upscaling rules in Sections 4.2 and 4.3.
4. Included 2DGS*-MCMC as an additional baseline in Tables 1 and 2 to better demonstrate the effectiveness of our texture upscaling strategy.
5. Added a texture decomposition analysis, along with corresponding figures, in Section 5.2.
6. Updated the **Limitations** and **Future Work** section for completeness.
7. Added render-speed evaluations and texture-size experiments in the Appendix.
8. Corrected various typos and minor inconsistencies throughout the paper.

We have highlighted all revisions in the updated manuscript. Below, we address the specific concerns raised by each reviewer.

---

### Author Response · Authors · 2025-12-03
**Summary of Prior Discussions**

Dear Area Chair,

Thanks again to all reviewers and AC for reviewing our submission. We note the recent reviewer-leak incident on OpenReview, and we sincerely appreciate your time and effort in contributing to the ICLR review process. To assist your evaluation, we provide below a summary of the previous discussions for your convenience.


Our paper proposes **Adaptive Anisotropic Textured Gaussians (A²TG)**, a textured Gaussian representation where each 2D Gaussian is equipped with an anisotropic texture whose resolution and aspect ratio are **adaptively chosen based on gradient signals**. We also note that the reviewers’ overall assessments are **consistently positive**. Across reviews, they repeatedly highlight that:


- The method is innovative and practical, clearly targeting memory redundancy in textured Gaussians (GiPV, QVPU).


- A²TG improves memory efficiency and visual fidelity over fixed-square textures and achieves superior or comparable rendering quality with reduced storage (GiPV, Cw5H, Q3gj).

- The paper is well-written, well-structured, and readable, with clear motivation and detailed explanations (QVPU, Cw5H, Q3gj).


Overall, the reviewers already see A²TG as a strong and well-presented contribution, and their main concerns were about clarification, additional experiments and rendering/training speed—which we have now addressed in detail.

***Reviewer GiPV*** questioned how textures are fetched from UV coordinates and how gradients propagate, raised concerns about rendering speed with non-uniform textures. In response, we now explicitly state that texture lookups use bilinear interpolation in UV space and clarify the gradient derivation. We also add explicit FPS and per-frame inference-time comparisons, showing that A²TG maintains **real-time performance** and is faster than Textured Gaussians* due to fewer texture parameters. Importantly, the reviewer **explicitly considered raising his/her score** if these issues were clarified and revised, which we have now done point-by-point.



***Reviewer QVPU*** requested additional experiments, clearer mathematical notation , and clarification of several claims and training details.  We have added experiments to include 2x2 texture resolutions and 1M Gaussian counts, showing that A$^2$TG maintains **comparable or better quality with lower memory** in these regimes. The mathematical exposition has been tightened.




***Reviewer Cw5H*** raised concerns that our gradient and anisotropy thresholds might require dataset-specific tuning, that iterative upscaling might add runtime overhead, and that occlusion interactions are only handled implicitly, and also noted several typos.
 We clarify that a single set of parameters is used across all datasets, with no dataset-specific tuning. The new runtime experiments (FPS and per-frame latency) show that A$^2$TG remains real-time. All reported typos and wording issues have been corrected.



***Reviewer Q3gj*** questioned how much our upscaling strategy contributes beyond MCMC-style optimization, asked for visualizations or analyses of the learned textures themselves, and requested more detail on threshold choices and the discrete texture grid design.  To better isolate our contribution, we strengthen the baselines (including 2DGS*-MCMC and memory-matched comparisons to Textured Gaussians*), showing that **A$^2$TG achieves a better memory–quality trade-off than simply adding more Gaussians or using fixed textures.** We additionally introduce a texture decomposition analysis: rendering with (i) full A²TG (SH + textures), (ii) textures removed (SH only), and (iii) SH removed (textures only). This reveals that textures capture high-frequency, view-consistent residuals, while SH encodes low-frequency structure and shading, directly addressing the request to understand what the textures themselves are learning.

Finally, as mentioned above, no reviewer followed up after our rebuttal, and did not adjust their score. Thus the current scores may under-reflect the clarifications and additional analyses we have provided. We hope this summary helps contextualize the discussion and the technical contributions of A$^2$TG when you form your recommendation.

Sincerely,
The Authors

---

### Meta-Review · Area_Chair_EuNR · 2026-01-04

**Summary:**

The reviewers’ concerns that informed the decision can be summarized as follows:

1. Reviewers requested clarification on how textures are queried using UV coordinates, how gradients propagate through texture sampling, and whether the mathematical derivations were correct and consistently presented.

2. Given the use of anisotropic and non-uniform texture resolutions, reviewers were concerned about rendering efficiency, CUDA implementation overhead, and whether real-time performance could be maintained.

3. Requests were made for additional experiments, including comparisons with different fixed texture resolutions (e.g., 2×2), larger numbers of Gaussians (up to 1M), and stronger or more carefully matched baselines.

4. Reviewers questioned whether anisotropy thresholds and gradient criteria required dataset-specific tuning and whether the upscaling rules were well justified.

5. Minor concerns were raised about missing discussion of texture downscaling, broader applicability, and potential extensions, which reviewers suggested should be acknowledged as limitations or future work.

**Reviewer Concerns:**

Concerns addressed by the rebuttal and revision:

1. Texture fetching and gradient correctness: The authors clarified that bilinear interpolation is used for texture sampling, revised the gradient derivations, fixed ambiguous notation, and ensured consistency across equations.

2. Rendering speed and efficiency:  New experiments demonstrate that A²TG achieves real-time performance (≈140 FPS with 500k Gaussians) and is faster than prior textured Gaussian baselines due to reduced texture parameter counts.

3. Additional experiments and baselines: The authors added results for 2×2 texture resolutions, experiments with up to 1M Gaussians, stronger and memory-matched baselines, and ablations isolating the effect of adaptive textures.

4. Hyperparameter robustness: The authors clarified that a single set of thresholds is used across all datasets, provided intuitive explanations for the anisotropic upscaling rules, and revised the text to remove ambiguities.

5. Presentation issues and typos:  Mathematical notation was clarified, contradictory claims were fixed, and reported typos and wording issues were corrected.

Remaining or partially outstanding concerns

Dynamic texture downscaling: Not implemented, but explicitly acknowledged and discussed as future work. This is a reasonable and non-blocking limitation.

**Reviewer Scores:**

Reviewer GiPV (initial score: 6):
Likely increase to 8. The reviewer explicitly stated they would be inclined to raise their score if texture sampling, gradient correctness, and rendering speed were clarified, all of which were addressed.

Reviewer QVPU (initial score: 6):
Likely increase to 8. Additional experiments (2×2 textures, 1M Gaussians), clarified notation, and runtime analysis directly addressed the reviewer’s key concerns.

Reviewer Cw5H  (initial score: 6)
Likely to be unchanged or increase to 8. Concerns are addressed.

Reviewer Q3gj (initial score: 4)
Likely increase to 6. These resppnse directly address the reviewer’s core concerns and strengthen the interpretation of the results.

---

### Decision · Program_Chairs · 2026-01-26

Accept (Poster)